

# A revision of tetrapod footprints from the late Carboniferous of the West Midlands, UK

Luke E. Meade, Andrew S. Jones and Richard J. Butler

School of Geography, Earth and Environmental Sciences, University of Birmingham, Birmingham, United Kingdom

## ABSTRACT

A series of sandstone slabs from Hamstead, Birmingham (West Midlands, UK), preserve an assemblage of tetrapod trackways and individual tracks from the Enville Member of the Salop Formation (late Carboniferous: late Moscovian–Kasimovian). This material has received limited previous study, despite being one of the few British sites to preserve Carboniferous tetrapod footprints. Here, we restudy and revise the taxonomy of this material, and document it using 3D models produced using photogrammetry. The assemblage is dominated by large tracks assigned to *Limnopus* isp., which were made by early amphibians (temnospondyls). A number of similar but smaller tracks are assigned to *Batrachichnus salamandroides* (also made by temnospondyls). *Dimetropus leisnerianus* (made by early synapsids) and *Dromopus lacertoides* (made by lizard-like sauropsids such as araeoscelids) are also present. This ichnofauna contrasts with a slightly stratigraphically older, more extensive and better-studied assemblage from Alveley (Shropshire), which is dominated by small amphibians with relatively rare reptiliomorphs, but which lacks *Dromopus* tracks. The presence of *Dromopus lacertoides* at Hamstead is consistent with the trend towards increasing aridity through the late Carboniferous. It is possible that the assemblage is the stratigraphically oldest occurrence of this important amniote ichnotaxon.

Corresponding authors
Luke E. Meade,
luke.edward.meade@gmail.com
Richard J. Butler,
r.butler.1@bham.ac.uk

## INTRODUCTION

In 1912, in a paper presented to the Geological Society of London, Walter Henry Hardaker described fossils, including a series of tetrapod footprints and trackways, from what were then considered Permian rocks near Birmingham in the British Midlands. Hardaker was a local school teacher and amateur field botanist who had studied at the nearby University of Birmingham (*Andrews, 1973*). He had discovered the footprints at New Quarry, in what was then the village of Hamstead, northwest of the city of Birmingham (Fig. 1). Today, the strata yielding these fossils fall within the metropolitan borough of Birmingham, but New Quarry itself has since been filled in. The fossils collected by Hardaker are housed in the collections of the University of Birmingham's Lapworth Museum of Geology (BIRUG) and the Birmingham City Museum and Art Gallery (BMAG).

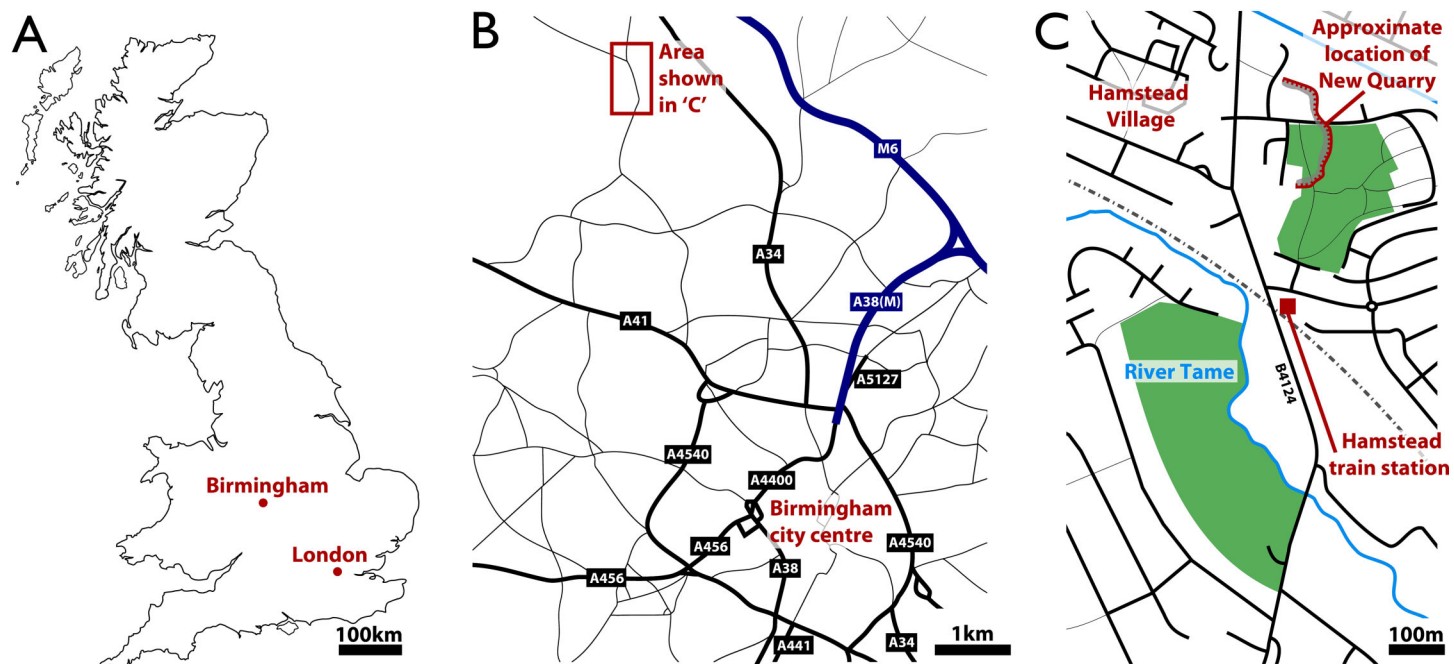

**Figure 1** **Geographic setting of Hamstead and the site of discovery of the fossil assemblage, 'New Quarry.'** (A) Location of Birmingham within Britain. (B) Location of Hamstead Village approximately 3 miles NW of Birmingham city centre. (C) Location of 'New Quarry' within the area of Hamstead.

*Hardaker (1912)* documented similarities between the tracks from Hamstead and those described by *Pabst (1908)* from the Upper Rotliegend Group (lower Permian) of Thuringia, Germany. Hardaker identified six footprint morphotypes that he assigned to different ichnospecies within Pabst's ichnogenus *Ichnium*. Hardaker used the similarities between the Hamstead and German footprints to argue for a lower Permian age for the beds exposed at Hamstead; however, subsequent stratigraphic work has assigned the Hamstead deposits to the Enville Member of the Salop Formation. The Salop Formation is dated as late Carboniferous on the basis of macrofloral remains (*Besly & Cleal, 1997*).

The taxonomy of the Hamstead footprints was partially revised by *Haubold & Sarjeant (1973)* and *Haubold & Sarjeant (1974)*. They reassigned five of the six footprint types that Hardaker had assigned to *Ichnium* to alternative ichnogenera. The ichnospecies *Ichniotherium cottae*, *Dimetropus leisnerianus* and *Dromopus lacertoides* were identified based upon re-examination of the original material whilst *Gilmoreichnus brachydactylus* and *Anthichnium salamandroides* were tentatively identified based upon Hardaker's illustrations. Subsequently, the Hamstead tracks were briefly revisited as part of a Ph.D. thesis by *Tucker (2003)*. She found no evidence for the presence of *Ichniotherium cottae*, but identified instead *Limnopus* (*Limnopus*) *vagus*, *Limnopus* (*Batrachichnus*) *salmandroides* (recombination of *Anthichnium salamandroides*), *Hyloidichnus bifurcatus* (reidentification of *Gilmoreichnus brachydactylus*), *Dimetropus leisnerianus*, and *Dromopus lacertoides*. However, she provided only one figure of the material, and did not discuss the rationale behind the revised identifications.

Tetrapod ichnofaunal localities are rare in the late Carboniferous of the UK. The most extensive and most thoroughly studied assemblage of late Carboniferous footprints comes from the Alveley Member of the Salop Formation at Alveley, southern Shropshire (*Haubold & Sarjeant, 1973*; *Haubold & Sarjeant, 1974*; *Tucker & Smith, 2004*). A small number of other localities in the Midlands and Somerset (*Haubold & Sarjeant, 1973*; *Haubold & Sarjeant, 1974*; *Milner, 1994*) have also yielded tetrapod footprints. Because of this scarcity, we here provide a redescription and reassessment of the Hamstead footprints and make comparisons to other late Carboniferous ichnofaunas, particularly that from Alveley. We discuss the implications for understanding tetrapod evolution through the late Carboniferous, an interval of major floral and faunal turnover dubbed the 'Carboniferous Rainforest Collapse' (*Sahney, Benton & Falcon-Lang, 2010*).

## GEOLOGICAL SETTING

The material from New Quarry, Hamstead, revised here was collected from sandstones in the upper part of the Enville Member (formerly referred to as the Enville Beds or Enville Formation) of the Salop Formation (Warwickshire Group). Reaching a thickness of 100–247 m (*Powell et al., 2000*) in the West Midlands, the Enville Member has been proposed to be of Westphalian D to Stephanian age within regional European stratigraphy (*Waters, Glover & Powell, 1994*; *Glover & Powell, 1996*; *Johnson, Glover & Turner, 1997*; *Besly & Cleal, 1997*; *Powell et al., 2000*), which correlates to the late Moscovian to Kasimovian of the global Carboniferous stratigraphy (*Gradstein, Ogg & Schmitz, 2012*). The Enville Member is the upper of the two members of the Salop Formation, the other being the Alveley Member (formerly Keele Formation), the source of the Alveley track assemblage (*Haubold & Sarjeant, 1973*; *Haubold & Sarjeant, 1974*; *Tucker & Smith, 2004*).

The Enville Member comprises interbedded red mudstones and red-brown fine–coarse grained, locally pebbly sandstones which are mostly sublitharenite (*Waters, Glover & Powell, 1994*; *Glover & Powell, 1996*; *Johnson, Glover & Turner, 1997*). Lenticular beds of conglomerate, the clasts of which consist mostly of Carboniferous limestone and chert, are also present. The depositional environment of the Enville Member is considered to have been a well-drained fluvially dominated plain (*Besly, 1988*), being a slightly more arid environment with more sporadic fluvial deposition than in the underlying Alveley Member (*Glover & Powell, 1996*).

Within New Quarry, *Hardaker (1912)* noted the strata to be predominantly obscurely bedded purple marls which alternated irregularly with massive beds of red and green calcareous sandstone, all of which dipped south-east at 3–4°. Many of these sandstone beds were described as lenticular, inconsistent in character, with an undulating and irregular surface. Hardaker also observed a bed of massive conglomerate of variable thickness at the top of the New Quarry succession, with an unconformity separating it from the underlying beds. The footprint-bearing slabs collected by Hardaker were mostly found as loose blocks of sandstone, with a few being sourced in situ on the lower surfaces of sandstone beds, and were all identified as derived from the 7.5 m of marls and sandstones (referred to by Hardaker as the 'New-Quarry Marl Sub-group') below the

massive conglomerate. The fossils appear to have come from multiple levels within this sequence, rather than a single bed.

## MATERIALS AND METHODOLOGY

The material from Hamstead consists of 15 whole slabs (containing specimens BIRUG BU5268–5271, BU5274, BU5277–BU5280, BU5282–BU5286, and BMAG 19/14) and four slabs that have each broken into two pieces (containing specimens BIRUG BU3294, BU5267, BU5272, BU5273, BU5275, BU5276, and BU5281). Specimen numbers with BIRUG BU prefixes used here refer to either individual tracks or to trackways (when preserved), rather than to slabs. The slabs themselves have a separate numbering system (see Supplemental Information for further details). Two of the whole slabs (BIRUG BU5287 and BU5288) only preserve arthropod tracks and were not restudied. All tracks are preserved as convex hyporelief on the undersurfaces of red sandstone slabs, and many slabs also include raindrop impressions and/or desiccation cracks.

High-resolution 3D models of the footprint slabs were created using photogrammetric modelling in order to document the trackways and identify new data on trackway dimensions and morphology. Each slab (or multiple slabs if clearly originally connected to one another) was artificially lit and 30–100 photographs were taken at a range of heights following circular paths around the specimen. Photographs were taken using a tripod-mounted digital SLR camera (Nikon D5100) with a fixed Nikon 50 mm lens. This ensured even coverage of the entire specimen.

The photographs were imported into the software package Agisoft Photoscan 1.2.4 (Professional Edition), which uses automated point picking and triangulation of point clouds to produce high-resolution 3D meshes. All meshes are available for download at https://dx.doi.org/10.5281/zenodo.154382. The meshes were exported as .ply files into the freeware software package CloudCompare 2.7, which was used to visualise the models. These included digital 3D reliefs with coloured contour intervals (methodology from *Romilio & Salisbury, 2014*), and also reliefs with steep gradients (i.e. the edges of a footprint) brightly highlighted. These relief images are also available at the above link.

A summary of the former and revised taxonomic identifications is presented in Table 1 and Supplemental Information.

## SYSTEMATIC PALAEOICHNOLOGY

*Limnopus* isp. *Marsh, 1894* (Figs. 2 and 4)

**Referred material:** BIRUG BU5267, manus and partial pes (Figs. 2A–2C); BIRUG BU5268, single pes; BIRUG BU5270, poorly preserved partial manus and partial pes; BIRUG BU5271, four pes prints and several associated poorly preserved manus prints, arranged in a poorly preserved trackway; BIRUG BU5272, partial pes; BIRUG BU5278, manus and partial pes (Fig. 4); BIRUG BU5284, manus-pes pair (*Haubold & Sarjeant, 1973*: Plate v, Figs. 1, 2D and 2E); BIRUG BU5286, single manus; BMAG 19/14, single pes.

**Description:** Tetradactyl and plantigrade to semi-digitigrade manus varying greatly in size and reaching relatively large size for the ichnogenus. Manus width (45–121 mm) always

**Table 1 Former identifications of the Hamstead material alongside the revised taxonomy from this study and the associated track makers.**

| Hardaker's track type numbers | Hardaker (1912) taxonomy | Haubold & Sarjeant (1973) taxonomy | Tucker (2003) taxonomy | Revised taxonomy– this study | Trackmaker |
|---|---|---|---|---|---|
| H 1 | Ichnium sphaerodactylum | Ichniotherium cottae | Limnopus vagus | Limnopus isp. | Temnospondyl amphibian |
| H 1a | Ichniotherium cottae | Ichniotherium cottae | Limnopus vagus | Limnopus isp. | Temnospondyl amphibian |
| H 1b | Ichnium sphaerodactylum minus | Not restudied | Limnopus vagus | Limnopus isp. | Temnospondyl amphibian |
| H 2 | Ichnium pachydactylum | Not restudied | Limnopus vagus | Limnopus isp. | Temnospondyl amphibian |
| H 2a | Ichnium pachydactylum minus | Not restudied | Limnopus vagus | Limnopus isp. | Temnospondyl amphibian |
| H 2b | Ichnium pachydactylum ungulatum | Dimetropus leisnerianus | Dimetropus leisnerianus | Dimetropus leisnerianus | Non-therapsid synapsid |
| H 3 | Ichnium brachydactylum | Gilmoreichnus brachydactylum | Hyloidichnus bifurcatus | Batrachichnus salamandroides | Temnospondyl amphibian |
| H 4 | Ichnium doliodactylum | Anthichnium salamandroides | Limnopus salamandroides | Batrachichnus salamandroides | Temnospondyl amphibian |
| H 5 | Ichnium gampsodactylum | Dromopus lacertoides | Dromopus lacertoides | Dromopus lacertoides | Sauropsida |
| H 5a | Ichnium gampsodactylum minus | Dromopus lacertoides | Dromopus lacertoides | Dromopus lacertoides | Sauropsida |
| H 6 | Ichnium aerodactylum | Not restudied | Not restudied | Batrachichnus salamandroides | Temnospondyl amphibian |

greater than length (27–90 mm) where both can be measured with confidence. Digits are distinct, broad, and short with rounded ends. Digit III is the longest (16–35 mm), digit I the shortest (13–24 mm), with II and IV being intermediate and similar in length (17–30 and 17–29 mm, respectively). Manus is often turned inwards slightly and digits I–III sometimes curve forward on inwardly turned examples; this is especially the case in larger tracks. No curvature is found in digit IV. Digits and their corresponding area of the sole become increasingly deeply impressed from digit I to digit IV. There is no strongly impressed heel print. In some cases this is due to the manus print being overlapped by the digits of the pes, possibly obscuring any heel impression that may otherwise have formed.

The pes is pentadactyl and plantigrade, varying greatly in size as with the manus. Pes width (42–105 mm) is generally greater than length (21–112 mm) where such measurements are possible, apart from in the larger specimens in which length is typically greater than width. Digits are broad, longer than those of the manus and often lack any impression at the tips of digits III–V, especially in the case of larger tracks. Digit IV is the longest (13–61 mm) followed by digit III (12–47 mm). Digits II and V are intermediate and similar in length (10–36 and 12–39 mm, respectively) whilst digit I is the shortest (6–24 mm). Digits III and IV may exhibit a slight outward curvature, whereas the others are straight. Digits I and II have the strongest impressions, and the impressions decrease from digit III to digit V. The pes occasionally has a slight outward angulation.

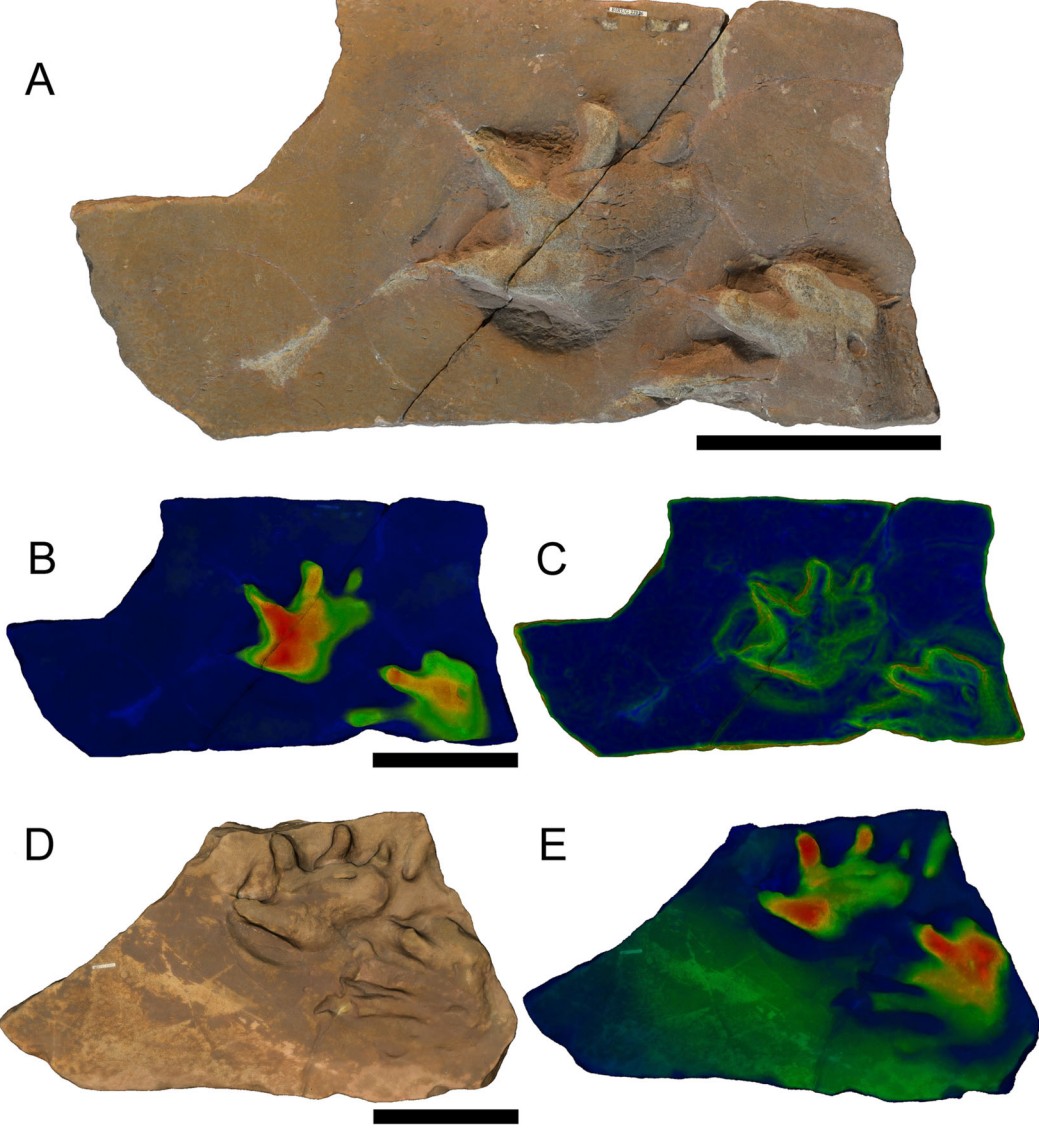

**Figure 2 (A–C) Specimens BIRUG BU5267 and (D–E) BIRUG BU5284, well defined examples of *Limnopus* isp. tracks from Hamstead.** All scale bars 10 cm. (A) Top-down photograph. (B) Tracks rendered to show relief with an arbitrary scale. (C) Tracks rendered to highlight areas of steep gradient, digitally isolating the outline of the tracks. (D) Digital rendering (photogrammetric model). (E) Tracks rendered to show relief with an arbitrary scale.

A heel is more strongly impressed than is the case with the manus but is still faint and often mostly absent.

Manus-pes overlap does occur but it is infrequent, slight, and difficult to judge due to the poor impression of the heel of the manus. The placement of the pes is posterior to and tends to be slightly lateral of the manus, with the exception of BIRUG BU5271 in which overstepping of the manus by the pes appears to be present (see below). Due to the fact that the described footprints are almost entirely isolated examples or from incomplete trackways, information on the morphology of the trackways is scarce. Available trackway measurements for BIRUG BU5271 are as follows: manus-pes separation 36 mm; manus stride 123 mm;

manus stride: footlength 4.6; pes stride: 105 mm; pes pace 65 mm; pes trackwidth 50 mm. A solitary large manus-pes pair measured 114 mm for manus-pes separation.

**Discussion:** The classification of this material using open nomenclature at the ichnospecies level is a result of the limited material available and the generally isolated nature of the footprints. The tracks have previously been assigned to *L. vagus*, an ichnospecies that tends to be approximately a third of the size of the largest tracks found amongst the Hamstead material (*Tucker & Smith, 2004*). Many previously erected ichnospecies within the *Limnopus* ichnogenus are now considered synonymous with *L. vagus*, but assigning the Hamstead specimens to *L. vagus* or any other ichnospecies is difficult without more complete trackways. Such large tracks of *Limnopus* are not unheard of; indeed unpublished tracks have been measured at over 200 mm in length (*Voigt & Haubold, 2015*).

The referral of BIRUG BU5271 to this ichnogenus is tentative. Although generally consistent in morphology with the other examples of *Limnopus*, the tracks in BIRUG BU5271 are considerably smaller, and are the only examples known from Hamstead to show apparent overstepping of the manus by the pes.

*Limnopus* tracks are known from all over Europe, with examples known from Britain (*Tucker & Smith, 2004*), Germany (*Haubold, 1971*; *Haubold, 1996*; *Voigt, 2005*), Spain (*Voigt & Haubold, 2015*), France (*Gand & Durand, 2006*), Poland (*Ptaszyński & Niedźwiedzki, 2004*; *Niedzwiedzki, 2015*), and Italy (*Marchetti, Avanzini & Conti, 2013*; *Marchetti et al., 2015*). They are also known from the USA (*Lucas et al., 2011*; *Lucas & Dalman, 2013*), Canada (*Van Allen, Calder & Hunt, 2005*), Argentina (*Hunt & Lucas, 1998*) and Morocco (*Voigt et al., 2011a*; *Voigt et al., 2011b*). They appear to range from the middle Moscovian to the middle Kungurian (*Tucker & Smith, 2004*; *Voigt & Lucas, 2015*). The trackmakers are considered to have been medium-large temnospondyl amphibians (*Haubold, 1971*; *Gand, 1987*; *Haubold, 2000*; *Tucker & Smith, 2004*; *Voigt, 2005*; *Gand & Durand, 2006*; *Voigt & Haubold, 2015*).

*Batrachichnus salamandroides* Geinitz, 1861 (Fig. 3)

**Referred material:** BIRUG BU3294, set of multiple extremely tiny prints; BIRUG BU5280, approximately eight very poorly preserved prints representing a single trackway; BIRUG BU5285, relatively well preserved trackway of ten manus-pes pairs and a tail drag.

**Description:** Tetradactyl and plantigrade manus of small size. Manus width (6–14 mm) is generally greater than manus length (7–13 mm). Digits of the manus are generally straight with an occasional inward curvature in digit III. Digit proportions are hard to determine and vary throughout the trackway though it appears digit I is shortest in most cases (1–4 mm) and digit II longest (3–5 mm). Digits III and IV both range from 2–4 mm in length. Manus is less strongly impressed than the pes though the whole foot is often represented, showing a gently rounded heel. This is obscured by an overlapping pes in two cases. Manus appears slightly more deeply impressed towards the digits.

Pes is pentadactyl and also plantigrade. Pes width (10–16 mm) is generally greater than pes length (8–15 mm). Digits are straight and show a slight distal taper in some cases.

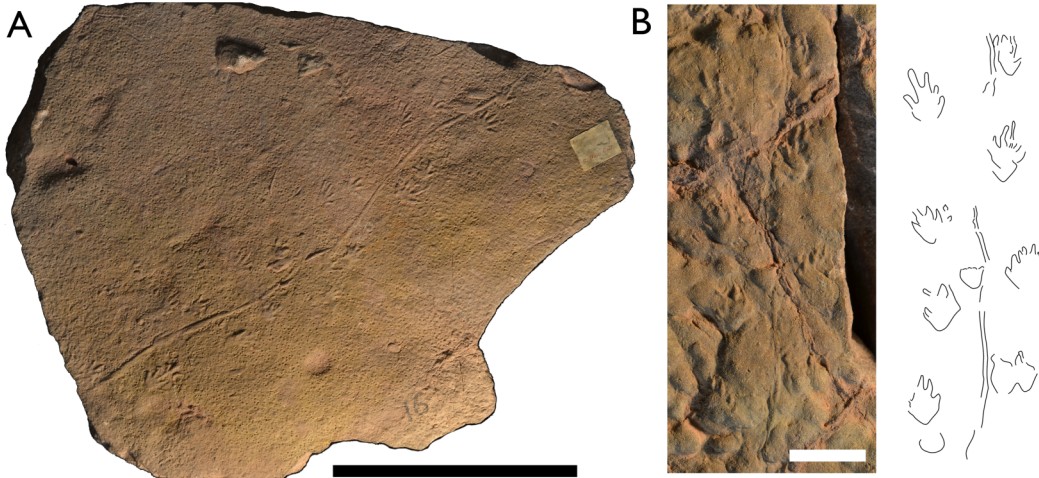

**Figure 3** *Batrachichnus salamandroides* **tracks from Hamstead.** Tracks were too weakly impressed to create a useful photogrammetric model. (A) Top-down photograph of specimen BIRUG BU5285, the most complete trackway of *B. salamandroides* amongst the material. Scale bar 10 cm. (B) Tiny *B. salamandroides* trackway from specimen BIRUG BU3294 accompanied by outline drawing. Scale bar 1 cm.

Though digit I is clearly the shortest (2–3 mm) the rest of the proportions are unclear and vary. Digit III is the longest on average (4–6 mm) followed by digit IV (3–6 mm). Digits II and V are similar in length on average (3–4 and 2–5 mm, respectively). The pes is more strongly impressed than the manus showing a rounded heel. Pes appears more deeply impressed towards the inner digits.

Manus is sometimes rotated inward, although this varies greatly within the trackway. A clear and continuous tail drag is present throughout. Manus stride 40–51 mm, manus pace 25–32 mm, pes stride 35–48 mm, pes pace 29–35 mm, manus-pes separation 6–17 mm. Pes track width (26 mm) slightly exceeds that of the manus (19 mm).

**Discussion:** These small tracks with a tetradactyl manus and tail drag match the characteristics of *Batrachichnus salamandroides*. It is considered the only valid ichnospecies within the genus *Batrachichnus* (*Tucker & Smith, 2004*; *Voigt, 2005*). Within the material assigned to *B. salamandroides* are a set of extremely small tracks (BIRUG BU3294) each no more than 4 mm in length. Their tiny size and poor preservation prevents accurate measurement but nevertheless their size, track morphology, and the presence of a tail drag are consistent with *B. salamandroides* and similar to the tracks reported by *Stimson, Lucas & Melanson (2012)*.

The larger *B. salamandroides* tracks amongst the Hamstead material are very similar in appearance to those assigned to *Limnopus* isp. They do however have relatively longer digits on the manus, a characteristic of *Batrachichnus* (*Haubold, 1996*; *Voigt, 2005*). The similarity between *Batrachichnus* and *Limnopus* is unsurprising as there has been much debate on the taxonomic relationship between the two ichnogenera (*Haubold, 1996*; *Haubold, 2000*; *Tucker & Smith, 2004*; *Voigt, 2005*; *Lucas et al., 2011*; *Voigt et al., 2011a*; *Voigt et al., 2011b*), with *Batrachichnus* possibly representing a juvenile form of *Limnopus*. It has been suggested that *Batrachichnus* should be reassigned as a subgenus of *Limnopus* (*Tucker & Smith, 2004*) but this has generally been rejected by subsequent

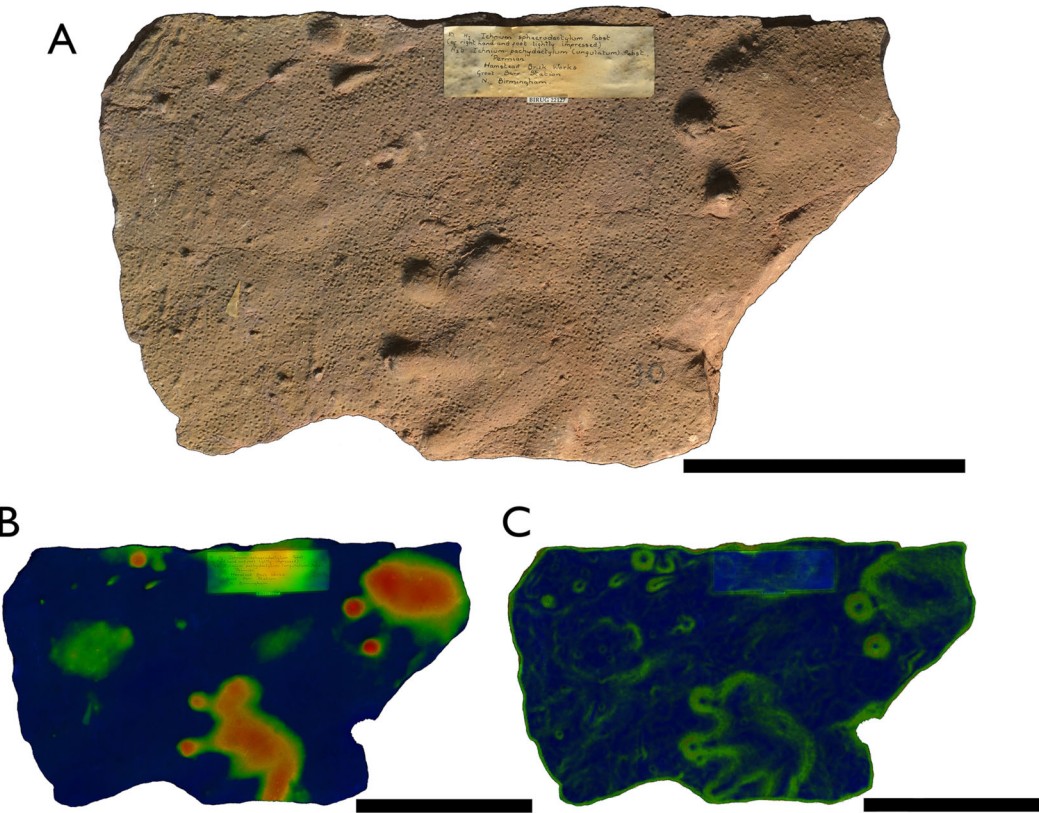

**Figure 4** Specimens BIRUG BU5277 (left surface of slab) and BIRUG BU5278 (right surface of slab), a *Dimetropus leisnerianus* pes print and *Limnopus* isp. manus-pes pair respectively from Hamstead. (A) Top-down photograph. (B) Tracks rendered to show relief with an arbitrary scale. (C) Tracks are rendered to highlight areas of steep gradient, digitally isolating the outline of the tracks. All scale bars 10 cm.

workers (*Lucas et al., 2011*). The relationship between the two remains questionable and the incomplete nature of the Hamstead trackways does not help the issue.

*Batrachichnus* is a very common ichnotaxon within Carboniferous and Permian assemblages, ranging from middle Moscovian to the Changhsingian (*Tucker & Smith, 2004*; *Voigt & Lucas, 2015*), with examples known from Britain (*Tucker & Smith, 2004*), Spain (*Voigt & Haubold, 2015*), France (*Gand & Durand, 2006*), Germany (*Voigt, 2005*), Poland (*Ptaszyński & Niedźwiedzki, 2004*; *Niedzwiedzki, 2015*), and Italy (*Avanzini et al., 2008*; *Marchetti, Avanzini & Conti, 2013*). It has also been found in the USA (*Lucas et al., 2011*; *Voigt, Lucas & Krainer, 2013*; *Voigt & Lucas, 2015*), Canada (*Stimson, Lucas & Melanson, 2012*), Argentina (*Melchor & Sarjeant, 2004*), and Morocco (*Voigt et al., 2011a*; *Voigt et al., 2011b*). *Batrachichnus* tracks are considered to have been made by small temnospondyl amphibians (*Haubold, 1971*; *Haubold, 1996*; *Haubold, 2000*; *Tucker & Smith, 2004*; *Voigt, 2005*).

*Dimetropus leisnerianus* Geinitz, 1863 (Fig. 4)

**Referred material:** BIRUG BU5269, three very poorly preserved tracks; BIRUG BU5275, three very poorly preserved tracks; BIRUG BU5277, single pes; BIRUG BU5279, manus and pes pair.
**Description:** Semi-digitigrade and pentadactyl manus. Only one faintly impressed example missing most of digits IV and V exists amongst the material, approximately 47 mm in length and at least 59 mm in width. Digits are not impressed along their whole length; generally only rounded, hemispherical impressions at the front of the sole representative of the metatarsal-phalangeal pads of the manus, and relatively deeply impressed claw marks are preserved. The impression of the claw for digit I is rotated laterally away from the others which face forward. Digit I is shorter than II and III which are very similar in length. The anteroposteriorly elongated sole and heel typical of *Dimetropus* (*Romer & Price, 1940*; *Voigt & Ganzelewski, 2010*; *Voigt et al., 2011a*; *Voigt et al., 2011b*; *Voigt & Lucas, 2015*) is not preserved amongst the material.

Pentadactyl and plantigrade pes. Pes width (81–95 mm) is greater than length (62–70 mm) and both dimensions are greater than those of the manus. Digit IV is the longest (23–26 mm) followed by digit V (21–23 mm), digit III (20 mm), and digit II (14–20 mm), with digit I being the shortest (10–21 mm). The impressions are weak; digits are not well preserved and are generally represented by hemi-spherical impressions of the metatarsal-phalangeal pads and claw marks, as in the manus. No elongated sole or heel is preserved in the pes.

Very little information can be identified or measured pertaining to trackways other than a manus-pes separation of 79 mm. There is no evidence of a tail drag.

**Discussion:** The tracks assigned to *Dimetropus leisnerianus* are generally quite poorly preserved, which is not uncommon (*Voigt & Lucas, 2015*), yet the tracks still show rather well the hemispherical impressions of the metatarsal-phalangeal pads and deep claw marks characteristic of *Dimetropus*. BIRUG BU5269 and BIRUG BU5275 are extremely poorly preserved, being deeply but morphologically poorly defined likely as a result of the substrate. In the case of an isolated track, such as BIRUG BU5277, manus and pes can be discerned by the relative length of digit V (*Voigt, 2005*). In the manus, the length of digit V corresponds with that of digit II. In the pes, the length of digit V corresponds instead that of with digit III. The latter is the case in BIRUG BU5277; thus it is identified as a pes track.

The separation of *Dimetropus* into ichnospecies beyond that of the type species, *D. leisneranus*, has been considered unsupported (*Voigt, 2005*; *Voigt, 2007*; *Voigt & Lucas, 2015*) although a new ichnospecies, *Dimetropus osageorum*, has been recently proposed (*Sacchi et al., 2014*). *Dimetropus osageorum* tracks exhibit proportionally shorter digits and a greater degree of heteropody than *D. leisnerianus*. This is not seen in the tracks from Hamstead which match the characteristics of *D. leisnerianus* well and thus the material is assigned to that ichnospecies here.

*Dimetropus* is well known from Western and Central Europe (*Tucker & Smith, 2004*; *Voigt, 2005*; *Voigt & Ganzelewski, 2010*), Eastern Europe (*Lucas, Lozovsky & Shishkin, 1999*; *Ptaszyński & Niedźwiedzki, 2004*; *Niedźwiedzki & Bojanowski, 2012*; *Voigt et al., 2012*), North America (*Hunt, Lucas & Lockley, 2004*; *Sacchi et al., 2014*; *Voigt & Lucas, 2015*), Morocco (*Voigt et al., 2011a*; *Voigt et al., 2011b*), and Argentina (*Hunt & Lucas, 1998*). *Dimetropus* tracks have been found from the middle Moscovian to the middle

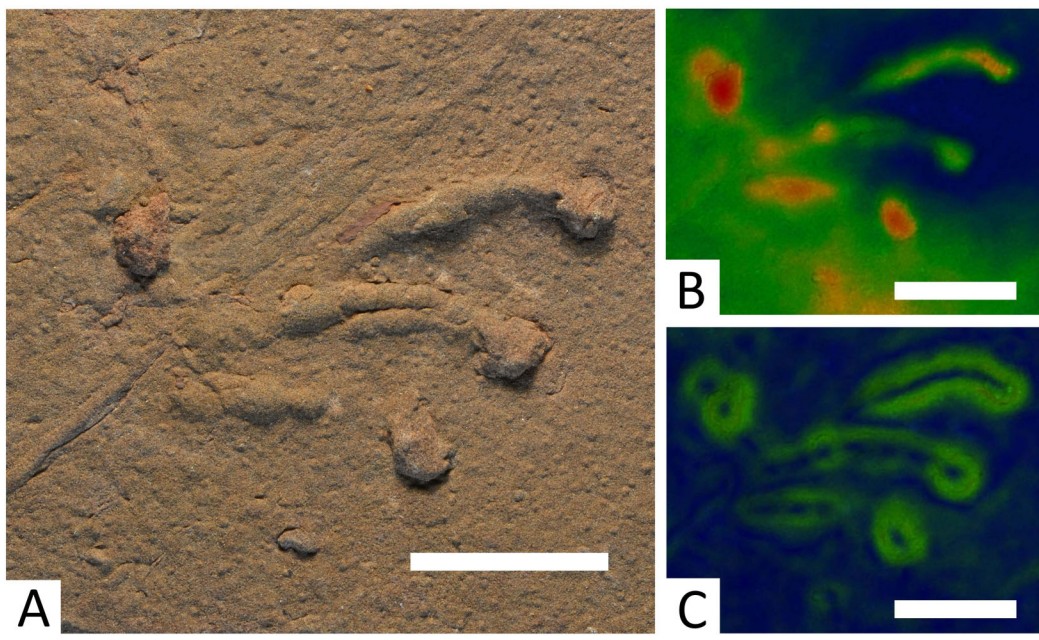

**Figure 5 Specimen BIRUG BU5283, only well-defined example of a *Dromopus lacertoides* track from Hamstead.** All scale bars 2 cm. (A) Top-down photograph. (B) Track rendered to show relief with an arbitrary scale. (C) Track rendered to highlight areas of steep gradient, digitally isolating the outline of the track.

Kungurian (*Tucker & Smith, 2004*; *Voigt & Lucas, 2015*). The trackmakers are considered to have been non-therapsid synapsids ('pelycosaurs') such as sphenacodonts, caseids, edaphosaurids, and ophiacodontids (*Haubold, 2000*; *Voigt, 2005*; *Voigt & Ganzelewski, 2010*; *Romano, Citton & Nicosia, 2016*).

*Dromopus lacertoides Geinitz, 1861* (Fig. 5)

**Referred material:** BIRUG BU5281, mostly indiscernible mass of tracks (*Hardaker, 1912*: Fig. 26; *Haubold & Sarjeant, 1973*: Plate 6, Fig. 1); BIRUG BU5282, single track (Fig. 5).

**Description:** The surface of BIRUG BU5281 is covered with a mass of tracks in which no clear trackways can be identified. The absence of a trackway prevents manus and pes from being distinguished, as they are morphologically almost identical in this ichnogenus (*Voigt, 2005*). These tracks are digitigrade and pentadactyl. All tracks are either incomplete or poorly preserved, preventing accurate measurement of length and width. Digit IV appears the longest (9–21 mm), followed by digit III (5–14 mm), digit II (2–10 mm), digit I (2–8 mm), then digit V (3 mm). Digits II–IV curve inwards and all digits are elongate and slender.

BIRUG BU5282 exhibits a single well-preserved track. The print is pentadactyl and digitigrade, length (48 mm) is greater than width (43 mm). Digit IV is the longest (43 mm), followed by digit III (34 mm), digit II (30 mm), digit I (19 mm), then digit V (10 mm). Digits II–IV curve inwards, whilst digit V is clearly directed out from the manus laterally and curves slightly towards the rear. All digits are long, slender, and end in a more deeply impressed circular tip.

**Discussion:** There appear to be many tracks of this type amongst the Hamstead material, but few are measurable: the vast majority occur within specimen BIRUG BU5281 as a confused, poorly preserved, and densely packed and frequently overlapping assemblage. A differentiation of species within the ichnogenus *Dromopus* beyond that of the type species has not yet been convincingly produced (*Gand, 1987*; *Haubold, 1996*; *Haubold, 2000*; *Haubold & Lucas, 2003*; *Voigt, 2005*; *Voigt & Haubold, 2015*; *Voigt & Lucas, 2015*). As a result, these tracks among the Hamstead material are assigned to *D. lacertoides*, the only currently valid ichnospecies, with which they are morphologically consistent.

*Dromopus* is a very abundant ichnogenus, the most common and widespread tetrapod track of the late Palaeozoic (*Haubold, 1971*; *Haubold, 1996*; *Haubold, 2000*; *Gand, 1987*; *Voigt, 2005*; *Voigt, 2007*; *Voigt, 2012*; *Lucas & Hunt, 2006*; *Lucas, 2007*), being known within Europe in Germany (*Haubold, 1971*; *Voigt, 2005*), France (*Gand, 1987*; *Gand & Durand, 2006*), Spain (*Voigt & Haubold, 2015*), Italy (*Avanzini et al., 2008*), Russia (*Lucas, Lozovsky & Shishkin, 1999*), and Poland (*Voigt et al., 2012*). It has also been reported from the USA (*Lucas et al., 2011*; *Voigt, Lucas & Krainer, 2013*; *Voigt & Lucas, 2015*), Canada (*Van Allen, Calder & Hunt, 2005*), and Morocco (*Voigt et al., 2011a*; *Voigt et al., 2011b*). *Dromopus* tracks elsewhere range from the late Moscovian to the Changhsingian (*Tucker & Smith, 2004*; *Voigt & Lucas, 2015*). The Enville Member which bears the Hamstead assemblage has been dated as late Moscovian or Kasimovian on the basis of macroflora. It is therefore possible that the Hamstead material is one of the stratigraphically earliest occurrences of the *Dromopus* ichnogenus. Tracks of this type are thought to have been produced by lizard-like sauropsids (eureptiles and parareptiles) of small–medium size such as bolosaurids and araeoscelids (*Haubold, 1971*; *Haubold, 1996*; *Haubold, 2000*; *Gand, 1987*; *Haubold & Lucas, 2003*; *Voigt, 2005*; *Gand & Durand, 2006*).

Tetrapoda indet.

**Referred material:** BIRUG BU5273, two manus-pes pairs and an isolated manus forming a trackway; BIRUG BU5274, indiscernible mass of prints; BIRUG BU5276, approximately six very poorly preserved prints; BIRUG BU5283, partial pes.

**Discussion:** a number of tracks within the Hamstead material are too poorly preserved to identify to ichnogenus. Those of BIRUG BU5273 resemble those assigned to both *Limnopus* and *Batrachichnus salamandroides*, but are intermediate in size between the two. As such we leave them unassigned at the generic level. The following trackway measurements are possible for BIRUG BU5273: manus-pes separation 38 and 46 mm; manus stride 116 mm; manus pace 76 and 69 mm; manus stride: footlength 5.9; pes pace 92 mm; manus trackwidth 40 mm; pes trackwidth 62 mm.

## DISCUSSION

The late Carboniferous was an interval of major environmental change on a global scale. Increasing aridity during the late Moscovian and the Kasimovian led to the

**Table 2 Tetrapod ichnotaxa and their supposed trackmakers reported from Alveley (*Tucker & Smith, 2004*) contrasted with those from Hamstead.**

| Ichnospecies | Trackmaker | Alveley | Hamstead |
| --- | --- | --- | --- |
| *Batrachichnus salamandroides* | Temnospondyl amphibian | X | X |
| *Dimetropus leisnerianus* | Non-therapsid synapsid | X | X |
| *Dromopus lactertoides* | Sauropsida | | X |
| *Hyloidichnus bifurcatus* | Captorhinomorphs | X | |
| *Ichniotherium willsi* | Diadectomorphs | X | |
| *Limnopus* isp. | Temnospondyl amphibian | | X |
| *Limnopus plainvillensis* | Temnospondyl amphibian | X | |
| *Limnopus vagus* | Temnospondyl amphibian | X | |

collapse and fragmentation of the widespread, humid, tropical rainforests (the 'Coal Forests') that were so typical of the Carboniferous period. This has been recognised as one of the two key mass extinction events in the plant fossil record (*Cascales-Miñana & Cleal, 2014*) and was likely the driving force of significant changes in terrestrial tetrapod communities (*Sahney, Benton & Falcon-Lang, 2010*). Amphibians, which dominated ecosystems during the Carboniferous, declined in diversity, whereas early amniotes appear to have been largely unaffected and continued to diversify (*Sahney, Benton & Falcon-Lang, 2010*). The success of amniotes has been linked to their greater independence from water, potentially providing the group with an ecological advantage in the new, drier conditions. The footprint assemblages of the Salop Formation from Alveley (Alveley Member) and Hamstead (Enville Member) potentially capture key phases of this faunal transition, dating as they do to the late Moscovian (Westphalian D) and late Moscovian–Kasimovian (Westphalian D–Stephanian), respectively.

The Hamstead assemblage has yielded four classic late Carboniferous–Early Permian ichnotaxa: *Limnopus* isp., *Batrachichnus salamandroides*, *Dimetropus leisnerianus*, and *Dromopus lacertoides*, representing temnospondyl amphibians of a range of different sizes, early synapsids (pelycosaurs), and sauropsid reptiles. The stratigraphically older tracks from Alveley include six ichnospecies: *Ichniotherium willsi*, *Hyloidichnus bifurcatus* (albeit identified with some uncertainty on the basis of a single, poorly preserved specimen), *Dimetropus leisnerianus*, *Limnopus plainvillensis*, *Limnopus vagus* and *Batrachichnus salamandroides* (*Tucker & Smith, 2004*), representing the taxa present at Hamstead, as well as diadectomorphs (*Ichniotherium*) and possibly also captorhinomorphs (single track of ?*Hyloidichnus*) (*Haubold, 2000*; *Tucker & Smith, 2004*; *Voigt, Berman & Henrici, 2007*; *Voigt et al., 2010*). *Hyloidichnus bifurcatus* and *Ichniotherium willsi* are each rare in the Alveley material so their absence at Hamstead could feasibly be explained by the much smaller sample size known from the latter locality. It is also possible that over the time between the deposition of the Alveley Member and the Enville Member the trackmakers of *Ichniotherium willsi* and *Hyloidichnus bifurcatus* became locally (although not globally) extinct due to environmental changes. Differing environmental conditions could also be an explanation for the much greater size range of *Limnopus* tracks seen within the

Hamstead assemblage when compared to Alveley. Table 2 contrasts tetrapod ichnotaxa reported from Alveley (*Tucker & Smith, 2004*) and Hamstead alongside their supposed trackmakers.

One key difference between the assemblages from Alveley and Hamstead is the presence of *Dromopus lacertoides* at Hamstead, and its absence within the Alveley assemblage. The described ichnotaxa from Alveley are characteristic of marginal freshwater-terrestrial tetrapod communities (*Tucker & Smith, 2004*) whereas *Dromopus*, though also found in wet environments, is commonly associated with dune facies and more arid or inland environments, such as those with shallow lacustrine or playa-like settings (*Tucker & Smith, 2004*). Tetrapod ichnofaunal change through the late Carboniferous–early Permian has been observed in the USA, with assemblages becoming increasingly dominated by *Dromopus* tracks, whilst the tracks of temnospondyls (*Limnopus* and *Batrachichnus*) decline (*Lucas, Krainer & Voigt, 2013*). The Alveley and Hamstead assemblages likely document a similar ichnofaunal and environmental change.

## ACKNOWLEDGEMENTS

The authors would like to thank Kate Riddington and Jon Clatworthy (Lapworth Museum of Geology) for their help and for access to the specimens, and Sebastian Voigt for discussion. We thank Sebastian Voigt and Grzegorz Niedźwiedzki for review comments that improved the final manuscript, and Jérémy Anquetin for editorial suggestions.

### Funding

Funding was provided by a Palaeontological Association Undergraduate Research Bursary (PA-UB201503) to LEM. The funders had no role in study design, data collection and analysis, decision to publish, or preparation of the manuscript.

### Grant Disclosures

The following grant information was disclosed by the authors:
Palaeontological Association Undergraduate Research Bursary: PA-UB201503.

### Competing Interests

There are no competing interests.

### Author Contributions

- Luke E. Meade performed the experiments, analyzed the data, wrote the paper, prepared figures and/or tables, reviewed drafts of the paper.
- Andrew S. Jones performed the experiments, analyzed the data, wrote the paper, reviewed drafts of the paper.
- Richard J. Butler conceived and designed the experiments, wrote the paper, prepared figures and/or tables, reviewed drafts of the paper.

## Data Deposition

Zenodo: https://dx.doi.org/10.5281/zenodo.154382.

## Supplemental Information

Supplemental information for this article can be found online at http://dx.doi.org/10.7717/peerj.2718#supplemental-information.

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
