# Peer review of "A revision of tetrapod footprints from the late Carboniferous of the West Midlands, UK"

_PeerJ, doi:10.7717/peerj.2718_

## Round 0.1 · original submission · Minor Revisions

The two reviewers agree that your study is well written and executed. Overall, their comments are relatively minor and should be no trouble for you to respond to.

In addition to the reviewer comments, I have only noted a few very minor issues:

- l. 78: Milner (1994) not listed in the references at the end of the ms
- l. 99: why capitalize ‘limestone’ here?
- l. 184: too much space at the beginning of the sentence
- l. 188: remove ‘as’
- l. 277: which Voigt et al., 2011? a or b?
- l. 332: Voigt (2010) not listed in the references at the end of the ms (unless it is the Voigt 2012 that is not cited in the text; see next point)
- l. 529: Voigt 2012 not cited in the main text
- Figure 1: as suggested by the first reviewer, enlarge the fonts on this figure to improve readability

·

Basic reporting

This manuscript revises a hotherto poorly known tetrapod ichnofauna that is important regarding the evolution of early tetrapods, especially the early radiation of amniotes. Therefore, this work is of much interest to a broad community of vertebrate palaeontologists and evolutionary ecologists. The manuscript is sententiously written and meets the standards of the field.

Experimental design

The manuscript might benefit from another table contrasting tetrapod ichnotaxa reported from Alveley (Haubold & Sarjeant, 1973, 1974; Tucker & Smith, 2004) and Hamstead (this ms) with their supposed trackmakers.

Validity of the findings

I have to raise some points that that need attention or further explanation:
(1) Location labels in Fig. 1B and 1C are way too small. Please, fix!
(2) Dimetropus and Fig. 4: How did you discern manus and pes tracks in Dimetropus? This is not clear in isolated tracks as BIRUG 5277. I agree with the identification of BIRUG 5277 to be a pes imprint. There is, however, no explanation. To my experience isolated manus and pes tracks can only be discerned by the relative length of the fifth digit: manual digit V ~ manual digit II, pedal digit V ~ pedal digit III! You should state this in the description (maybe even with reference, e.g. Voigt, 2005, if not visible in the rather incomplete Hamstead track material). Anyway, BIRUG 5277 appears to be a pedal track according tot he relative length of pV.
(3) Line 295-296: I agree with the ichnospecies assignation, but you should state potential differences between D. leisnerianus and D. osageorum in order to avoid that people will think you judge by stratigraphic age only (D. leisnerianus older than D. osageorum)!
(4) Lines 311-321: I do not get how you are able to discern manus and pes among isolated Dromopus tracks. Dromopus shows manus and pes imprints of almost identical morphology. Thus, manus and pes tracks can only be discerned along trackways…
(5) Lines 376-383: I do not agree with the identification of Hyloidichnus from Alveley by Tucker & Smith (2004). It would be helpful to discuss thís identification of a single and poorly preserved specimen more carefully!
(6) Lines 383-386: Extinction is most unlikely with respect to Ichniotherium as this ichnotaxon has occurrences in rocks ranging in age from about 320 to 280 Ma.
(7) Lines 392 to 393: „…whereas Dromopus is an ichnotaxon generally associated with dune facies and more arid or inland environments“ – this is an unbased statement in my opinion. I agree that Dromopus also occurs in dune facies, more arid and even playa-like settings but it is within these facies not significantly more common than in wet environments. Hamstead might be the stratigraphically earliest occurrence of Dromopus – this is really important!

Additional comments

The Hamstead tetrapod ichnofauna might play a key role in restricting the the first occurrence of lacertoid tracks (Dromopus). Try to show up this more clearly and it will significantly add to the importance of this good work!

·

Basic reporting

I have reviewed the manuscript, “A revision of tetrapod footprints from the late Carboniferous of the West Midlands, UK" submitted to Peer J by Luke E Meade and co-authors. The manuscript is well written and addresses a very interesting and somewhat debated topic - the early amniote trace fossil record - provides ichnological evidence for the tetrapod diversity of the latest Carboniferous of UK (late Moscovian–Kasimovian). This report is of demonstrably great importance and broad interest. I don’t have any major comments or remarks to this manuscript.

Batrachichnus traces occur also in late Carboniferous and late Permian of Poland (see Ptaszyński & Niedźwiedzki, 2004; Niedźwiedzki, 2015).

Maybe the authors will want to add these references in the disscussion about record of Batrachichnus.

Ptaszyński, T. and Niedźwiedzki, G. 2004. Late Permian vertebrate tracks from the Tumlin Sandstone, Holy Cross Mountains, Poland. Acta Palaeontologica Polonica 49 (2): 289–320.

Niedzwiedzki, G., 2015: Carboniferous tetrapod footprints from the Lublin Basin, SE Poland. GFF, Vol. 137 (Pt. 1).

Experimental design

The authors have done considerable work investigating a historical collection. They studied several different features of the specimens (analyzed specimens in several ways and accurately described the many important morphological features of individual traces and provided a comprehensive comparison with other record). The manuscript in the present shape does not appear to have been published elsewhere previously. All figures are well prepared and useful for comparison.

Validity of the findings

The material is of particular interest because tetrapod footprints from late Carboniferous are somewhat rare. Clearly, the described findings demonstrate the presence of several tetrapods in the late Carboniferous of Hamstead, Birmingham. These trace fossils have a great potential to our understanding of Paleozoic tetrapod diversity in this part of Europe.

Additional comments

This study will certainly contribute to our understanding of the fossil tetrapod traces and also Carboniferous ecosystems. In general, the manuscript is very interesting and I would recommend publication in Peer J.

---

## Round 0.2 · accepted · Accept

Thank you for this straightforward revised version, which incorporates all suggestions made by the reviewers. I can now accept your manuscript for publication in PeerJ. Congratulations!